# Integrating AI into Cancer Immunotherapy—A Narrative Review of Current Applications and Future Directions

**DOI:** 10.3390/diseases13010024

**Published:** 2025-01-20

**Authors:** David B. Olawade, Aanuoluwapo Clement David-Olawade, Temitope Adereni, Eghosasere Egbon, Jennifer Teke, Stergios Boussios

**Affiliations:** 1Department of Allied and Public Health, School of Health, Sport and Bioscience, University of East London, London E16 2RD, UK; 2Department of Research and Innovation, Medway NHS Foundation Trust, Gillingham, Kent ME7 5NY, UK; j.teke@nhs.net; 3Department of Public Health, York St John University, London E14 2BA, UK; 4Endoscopy Unit, Glenfield Hospital, University Hospitals of Leicester NHS Trust, Leicester LE3 9QP, UK; aanuclement23@gmail.com; 5Department of Public Health, University of Dundee, Dundee DD1 4HN, UK; temitopeogundare2015@gmail.com; 6Department of Tissue Engineering and Regenerative Medicine, Faculty of Life Science Engineering, FH Technikum, 1200 Vienna, Austria; eghosaseregabriel@gmail.com; 7Department of Surgery, Medway NHS Foundation Trust, Gillingham, Kent ME7 5NY, UK; 8Faculty of Medicine, Health and Social Care, Canterbury Christ Church University, Canterbury, Kent CT1 1QU, UK; stergiosboussios@gmail.com; 9Faculty of Life Sciences & Medicine, School of Cancer & Pharmaceutical Sciences, King’s College London, Strand, London WC2R 2LS, UK; 10Kent Medway Medical School, University of Kent, Canterbury, Kent CT2 7LX, UK; 11AELIA Organization, 9th Km Thessaloniki—Thermi, 57001 Thessaloniki, Greece; 12Department of Medical Oncology, Medway NHS Foundation Trust, Gillingham, Kent ME7 5NY, UK

**Keywords:** artificial intelligence, cancer immunotherapy, biomarkers, predictive models, personalized medicine

## Abstract

Background: Cancer remains a leading cause of morbidity and mortality worldwide. Traditional treatments like chemotherapy and radiation often result in significant side effects and varied patient outcomes. Immunotherapy has emerged as a promising alternative, harnessing the immune system to target cancer cells. However, the complexity of immune responses and tumor heterogeneity challenges its effectiveness. Objective: This mini-narrative review explores the role of artificial intelligence [AI] in enhancing the efficacy of cancer immunotherapy, predicting patient responses, and discovering novel therapeutic targets. Methods: A comprehensive review of the literature was conducted, focusing on studies published between 2010 and 2024 that examined the application of AI in cancer immunotherapy. Databases such as PubMed, Google Scholar, and Web of Science were utilized, and articles were selected based on relevance to the topic. Results: AI has significantly contributed to identifying biomarkers that predict immunotherapy efficacy by analyzing genomic, transcriptomic, and proteomic data. It also optimizes combination therapies by predicting the most effective treatment protocols. AI-driven predictive models help assess patient response to immunotherapy, guiding clinical decision-making and minimizing side effects. Additionally, AI facilitates the discovery of novel therapeutic targets, such as neoantigens, enabling the development of personalized immunotherapies. Conclusions: AI holds immense potential in transforming cancer immunotherapy. However, challenges related to data privacy, algorithm transparency, and clinical integration must be addressed. Overcoming these hurdles will likely make AI a central component of future cancer immunotherapy, offering more personalized and effective treatments.

## 1. Introduction

Cancer remains one of the most significant challenges to global health, contributing to nearly 10 million deaths annually, making it the second-leading cause of death worldwide [1]. The global burden of cancer continues to escalate due to several factors, including an aging population, lifestyle changes, and the increasing prevalence of risk factors such as tobacco use, obesity, and physical inactivity [1]. This alarming trend highlights the urgent need for innovative cancer treatment strategies, as traditional methods, while effective in some cases, have significant limitations.

Traditional cancer treatments, such as surgery, chemotherapy, and radiation therapy, have been the mainstay of oncological care for decades [2]. However, they are not without significant drawbacks. Chemotherapy, for example, involves the use of cytotoxic drugs designed to target and kill rapidly dividing cells, a hallmark of cancer [3]. Unfortunately, this lack of specificity means that chemotherapy also affects healthy cells, leading to a range of toxicities, including gastrointestinal adverse events, fatigue, hair loss, and immunosuppression [3,4]. This broad impact on both malignant and non-malignant cells can significantly diminish a patient’s quality of life and, in some cases, limit the extent to which chemotherapy can be administered [3].

Additionally, chemotherapy’s efficacy is often compromised by the development of drug resistance [5]. Cancer cells can evolve mechanisms to evade the cytotoxic effects of chemotherapy, leading to a reduction in treatment effectiveness over time. This resistance can be intrinsic, present before treatment begins, or acquired after exposure to chemotherapy, resulting in relapses and disease progression [5,6]. The challenge of overcoming drug resistance remains a major obstacle in the effective management of many cancers [7]. Radiation therapy, another cornerstone of traditional cancer treatment, uses high-energy radiation to damage the DNA of cancer cells, thereby preventing them from replicating and spreading [8]. While radiation therapy can be more targeted than chemotherapy, it still poses risks to surrounding healthy tissues [8]. This can result in side effects such as skin damage, fatigue, and an increased risk of secondary cancers [8,9]. The precision of radiation therapy has improved with technological advancements, but challenges remain, especially when tumors are located near vital organs or in areas difficult to reach without affecting healthy tissue [8,10].

In recent years, immunotherapy has emerged as a promising alternative to these traditional treatments [11,12,13,14]. An effective anti-tumor immune response consists of several essential stages—the release of tumor antigens from damaged or dying cancer cells; the uptake and presentation of these antigens by dendritic cells and other antigen-presenting cells; the priming and activation of T cells; the trafficking, infiltration, and accumulation of T lymphocytes and natural killer [NK] cells; and the recognition and elimination of cancer cells by cytotoxic T lymphocytes and NK cells. This sequence provides a vital framework for understanding the mechanisms behind both response and resistance to immunotherapy [11,15]. Unlike chemotherapy and radiation, which directly target cancer cells or the tumor itself, immunotherapy harnesses the body’s immune system to recognize and destroy cancer cells [11,16]. This approach capitalizes on the natural ability of the immune system to distinguish between normal and abnormal cells, offering a more targeted and potentially less toxic form of treatment [11]. However, the application of immunotherapy is not without its complexities [17,18]. The human immune system is a highly intricate network of cells, molecules, and tissues that work together to defend the body against infections and other diseases [19]. In the context of cancer, the immune system’s role becomes even more complex. Cancer cells are adept at evading immune detection, often by exploiting regulatory pathways that suppress immune responses [20,21,22]. This ability to “hide” from the immune system is one of the primary challenges in developing effective immunotherapies.

Furthermore, tumors are not uniform entities, but they are composed of heterogeneous populations of cells with different genetic and molecular characteristics [21]. Whole-genome sequencing and whole-exome sequencing are vital tools in cancer research [23]. They contribute to the detection of predisposition genes, risk stratification, and the identification of rare single-nucleotide polymorphisms [23]. These technologies facilitate the discovery of associations between various syndromes and cancer, provide insights into the tumor microenvironment, and help identify previously unknown mutations that could be valuable for future personalized treatments [23]. This heterogeneity can lead to varying responses to immunotherapy within the same tumor, with some cells being effectively targeted while others persist and contribute to disease progression [20]. Additionally, the tumor microenvironment—a complex milieu of immune cells, blood vessels, and other components surrounding the tumor—can either support or hinder the immune response, further complicating the effectiveness of immunotherapy [20,21,24]. Despite these challenges, the potential of immunotherapy to transform cancer treatment is immense [25]. The development of immune checkpoint inhibitors, which block proteins that prevent the immune system from attacking cancer cells, has already led to significant advancements in the treatment of several cancers, including melanoma, lung cancer, and renal cell carcinoma [25,26,27,28]. However, these therapies are not universally effective, and a substantial proportion of patients do not respond to them or experience only partial responses [25,29].

As researchers continue to unravel the complexities of the immune system and its interactions with cancer, the integration of artificial intelligence [AI] has emerged as a transformative tool in the development of cancer immunotherapy. By analyzing vast datasets, AI can identify patterns, predict patient responses, and enable the creation of more personalized and effective treatment strategies [30]. This combination of immunotherapy and AI holds immense potential for advancing cancer care, offering the possibility of more precise, targeted, and successful interventions that improve both patient outcomes and quality of life.

Despite this promise, significant challenges remain. The global burden of cancer, coupled with the limitations of traditional treatments like chemotherapy and radiation, underscores the urgent need for innovative approaches such as immunotherapy [31]. However, the complexity of immune responses, tumor heterogeneity, and treatment resistance pose barriers to its efficacy [32]. To address the existing gaps in the literature, this review identifies a critical need for a comprehensive synthesis of AI’s applications in cancer immunotherapy that not only highlights its transformative potential but also addresses the practical challenges and future opportunities. While previous studies have explored specific aspects of AI in cancer care, such as biomarker discovery or patient stratification, many fail to provide a holistic view that integrates these advancements with the broader challenges of implementation, regulatory approval, and clinical integration. This review is novel in its approach as it combines an evaluation of AI’s current contributions with a forward-looking analysis of its role in advancing personalized immunotherapy. The key objectives of this review are: (1) to summarize the current applications of AI in enhancing the efficacy of cancer immunotherapy, (2) to identify challenges in integrating AI into clinical practice, and (3) to propose future directions for research and development in this rapidly evolving field. By bridging the gap between theoretical potential and practical implementation, this review aims to guide researchers, clinicians, and policymakers in leveraging AI to transform cancer immunotherapy into a more precise, efficient, and accessible treatment modality.

Figure 1 illustrates the multi-step process of integrating AI into cancer immunotherapy, emphasizing its transformative role in improving precision and efficiency. Data acquisition forms the foundation of the process, with input from various sources, including genomic sequences, patient medical records, immune profiling, and imaging data. These diverse datasets are analyzed, processed, and integrated into AI and machine-learning (ML) models. The models are trained to recognize patterns predictive of patient responses to immunotherapy, serving as a decision-support tool for clinicians. This personalized approach helps tailor treatment strategies to individual patients. Furthermore, the system incorporates a feedback loop by continuously monitoring patient progress and feeding response data back into the AI, enabling ongoing refinement and improvement of healthcare delivery. As illustrated, this iterative process enhances precision in cancer immunotherapy, leading to more targeted and efficient therapies.

## 2. Methods

This narrative review was conducted to explore the current applications, challenges, and potential future directions of AI in the field of cancer immunotherapy. The review aimed to synthesize existing literature and provide a comprehensive overview of the intersection between AI and immunotherapy, focusing on how AI can enhance the effectiveness of cancer treatments.

### 2.1. Literature Search Strategy

A comprehensive literature search was conducted across multiple databases, including PubMed, Google Scholar, Scopus, and Web of Science, to identify relevant studies, reviews, and reports published between January 2010 and July 2024. The time frame was selected because advancements in AI and its applications in cancer immunotherapy have significantly accelerated over the past decade, with notable breakthroughs occurring after 2010. Earlier studies were excluded to ensure the focus remained on recent and relevant developments in AI methodologies, and their integration with cancer immunotherapy. The search terms used included “Artificial Intelligence”, “AI”, “machine learning”, “deep learning”, “cancer immunotherapy”, “immune checkpoint inhibitors”, “biomarkers”, “treatment resistance”, “tumor heterogeneity”, and “personalized medicine”. Boolean operators (AND, OR) were employed to combine these keywords effectively.

Figure 2 illustrates the PRISMA flowchart detailing the selection process for the included studies. The initial search identified 360 articles across four databases: PubMed (120), Google Scholar (80), Scopus (100), and Web of Science (60). After screening, 268 articles were excluded based on predefined inclusion and exclusion criteria, leaving 92 articles for full-text review. Of these, 38 were further excluded after in-depth evaluation, resulting in 54 studies included in the final review.

### 2.2. Inclusion and Exclusion Criteria

The inclusion criteria for this review were as follows: (1) studies published in peer-reviewed journals; (2) articles discussing the application of AI in immunotherapy for cancer treatment; (3) reviews and meta-analyses that provided comprehensive overviews of relevant topics; and (4) studies focusing on the challenges and future directions of integrating AI with immunotherapy. Exclusion criteria included: (1) articles not available in English; (2) studies focusing solely on non-cancer-related immunotherapies; and (3) papers that lacked a clear discussion of AI’s role in immunotherapy.

### 2.3. Data Extraction and Synthesis

Articles that met the inclusion criteria were carefully reviewed, and relevant data were extracted, including study objectives, methodologies, key findings, and conclusions. Emphasis was placed on identifying patterns and trends in the application of AI to various aspects of immunotherapy, such as biomarker discovery, prediction of patient responses, optimization of combination therapies, and identification of novel therapeutic targets. The findings were synthesized to provide a cohesive narrative on the current state of AI in cancer immunotherapy, highlighting both successes and challenges.

Figure 3 illustrates the distribution of the 54 studies included in the review across various key subsections. The largest focus is on biomarker identification (12 papers), followed by predicting patient response (10 papers) and drug development (9 papers). Other areas, such as combination therapies (8 papers), chimeric antigen receptor T (CAR-T) cell therapy (7 papers), and monitoring treatment response and clinical trial optimization (4 papers each), are also represented.

### 2.4. Quality Assessment

The quality of the included studies was assessed based on their methodology, the robustness of the data presented, and the relevance of the findings to the review’s objectives. Studies that provided high-quality, reproducible results with clear implications for clinical practice were prioritized in the synthesis of the review. Figure 4 presents a visual representation of the distribution of machine- and deep-learning techniques across key areas of the review. Biomarker identification accounts for the largest share, reflecting its significant role in leveraging AI for personalized cancer immunotherapy. Other areas, including predicting patient response, drug development, combination therapies, CAR-T cell therapy, monitoring treatment response, and clinical trial optimization, also exhibit notable applications of these techniques.

## 3. The Role of AI in Cancer Immunotherapy

AI is increasingly being integrated into the field of cancer immunotherapy, offering unprecedented opportunities to enhance treatment efficacy, predict patient responses, and discover novel therapeutic targets [30]. By leveraging its ability to process and analyze vast and complex biological datasets, AI is addressing some of the key challenges that have traditionally limited the effectiveness of immunotherapy. As immunotherapy continues to evolve as a cornerstone of cancer treatment, AI is poised to play a crucial role in refining and optimizing these therapies, ultimately leading to more personalized and effective patient care. Table 1 provides a detailed overview of the various ways AI is being applied to enhance cancer immunotherapy, offering a clearer understanding of how AI can contribute to more effective, personalized cancer treatment strategies. The table includes descriptions, methodologies, specific examples, and the overall impact of each AI application, making it a valuable resource for understanding the scope and potential of AI in this field.

### 3.1. Enhancing Immunotherapy Efficacy

The efficacy of immunotherapy varies significantly among patients, largely due to the complex and heterogeneous nature of tumors [32,42]. AI has also provided critical insights into CAR-T cell therapy, particularly in enhancing the design and optimization of CAR constructs [43,44]. By leveraging advanced algorithms, AI enables the identification of optimal target antigens and the prediction of CAR-T cell performance, significantly improving both safety and efficacy [43,44]. Additionally, AI-driven models streamline the engineering of CAR constructs, reducing the time and costs associated with traditional trial-and-error approaches [45]. AI has emerged as a powerful tool for identifying biomarkers that predict the effectiveness of immunotherapy, which is crucial for personalizing treatment and improving outcomes [30]. Traditionally, the discovery of biomarkers has been a labor-intensive process, limited by the complexity of biological data and the need for sophisticated analytical methods. AI, particularly through ML and deep-learning (DL) algorithms, has revolutionized this process by enabling the rapid analysis of large-scale genomic, transcriptomic, and proteomic data to uncover novel biomarkers that might be overlooked using conventional methods [43].

One of the key applications of AI in this context is the identification of biomarkers associated with the response to immune checkpoint inhibitors (ICIs), such as Programmed Cell Death Protein 1 (PD-1)/Programmed Cell Death Ligand 1 (PD-L1) and Cytotoxic T-lymphocyte associated protein 4 (CTLA-4) inhibitors [33]. These biomarkers can include specific genetic mutations, patterns of gene expression, or characteristics of the tumor microenvironment [33,46]. For example, studies have shown that AI can effectively analyze tumor mutational burden (TMB) and microsatellite instability (MSI), both of which are associated with better responses to ICIs [47,48]. AI models can integrate these biomarkers into predictive algorithms, helping clinicians to stratify patients and select those who are most likely to benefit from immunotherapy [33]. Figure 5 highlights the role of AI in biomarker identification using genomic, transcriptomic, and proteomic data. The flowchart depicts how DNA (genomic), RNA (transcriptomic), and protein structures [proteomic] are processed by AI systems to identify key biomarkers, such as PD-1, TMB, and MSI. These biomarkers are critical in guiding personalized treatment decisions in the clinical stage. By leveraging AI’s ability to analyze complex datasets, Figure 5 demonstrates how AI-driven biomarker discovery bridges the gap between laboratory findings and clinical applications, improving the precision and efficacy of cancer immunotherapy.

In addition to identifying biomarkers, AI also plays a crucial role in optimizing combination therapies, which have shown great promise in enhancing the effectiveness of immunotherapy [30]. Immunotherapy, when used in combination with other treatments such as chemotherapy, radiation therapy, or targeted therapies, can produce synergistic effects that improve overall treatment outcomes [49,50]. However, determining the most effective combination of therapies for individual patients has traditionally been a challenging and time-consuming process. AI models can analyze clinical data from diverse sources to predict the most effective treatment combinations, considering the unique genetic and molecular characteristics of each patient’s tumor [41].

For example, AI-driven analyses have been used to predict the success of combining ICIs with chemotherapy or radiation therapy [51]. These models consider various factors, including the genetic profile of the tumor, the presence of specific immune cells in the tumor microenvironment, and the patient’s overall health status [51]. By doing so, AI can help clinicians identify the optimal treatment regimen for each patient, reducing the reliance on trial and error and minimizing the risk of exposing patients to ineffective or harmful therapies [41]. Moreover, AI has the potential to refine the timing and sequencing of combination therapies [52]. For instance, research has shown that the effectiveness of combining radiation therapy with immunotherapy can depend on the sequence in which the treatments are administered [53]. AI models can simulate different treatment scenarios and predict the most effective sequence and timing for each patient, thereby maximizing the therapeutic benefits while minimizing side effects [54].

### 3.2. Predicting Patient Response

One of the most significant challenges in cancer immunotherapy is the variability in patient responses [55]. While some patients experience dramatic and long-lasting remissions, others may see little-to-no benefit, and some may even suffer severe immune-related adverse events (irAEs) [55,56]. Accurately predicting which patients are likely to respond to immunotherapy—and which are at risk for adverse effects—is critical for optimizing treatment strategies and improving patient outcomes [56,57]. AI has been instrumental in developing predictive models that assess a patient’s likelihood of responding to immunotherapy, thereby enabling more personalized and effective treatment plans [58,59,60]. AI models for predicting patient response to immunotherapy often integrate data from a wide range of sources, including genomic profiles, imaging data, and electronic health records (EHRs) [33]. By analyzing this comprehensive dataset, AI can identify patterns and correlations that may not be apparent through traditional statistical methods [30,34]. For instance, ML algorithms have been used to predict responses to PD-1/PD-L1 inhibitors by analyzing a combination of genomic alterations, immune signatures, and clinical features [61]. A recent study demonstrated that a DL model could accurately predict patient response to PD-1 blockade therapy by analyzing pre-treatment histology slides [61]. This non-invasive approach could help identify patients who are most likely to benefit from these therapies, thereby guiding clinical decision-making and improving outcomes [61].

In addition to predicting treatment efficacy, AI models are also being used to anticipate potential side effects, particularly irAEs, which can be severe and life-threatening. AI-driven tools have been developed to predict the likelihood of irAEs based on patient-specific factors, allowing clinicians to take proactive measures to mitigate risks [41]. For example, an AI model was able to predict the occurrence of irAEs in patients undergoing ICIs therapy by analyzing baseline immune-related biomarkers and clinical data [41]. This predictive capability provides a framework for personalized monitoring and management of side effects, which is critical for ensuring patient safety during treatment. Furthermore, AI plays a crucial role in refining patient selection for clinical trials involving immunotherapy [48]. By predicting patient responses more accurately, AI can help identify appropriate candidates for specific trials, increasing the likelihood of successful outcomes and accelerating the development of new therapies [62]. For instance, AI algorithms have been used to screen patients for trials based on their predicted response to specific immunotherapies, thereby improving the efficiency and effectiveness of the trial process [40].

### 3.3. Discovering Novel Therapeutic Targets

The discovery of novel therapeutic targets is essential for the continued development of effective cancer immunotherapies [63]. AI has emerged as a powerful tool for identifying new targets that may not be apparent through traditional research methods [64]. By analyzing vast datasets, including genetic, epigenetic, and transcriptomic data, AI can identify potential targets that could lead to the development of new immunotherapeutic agents, thereby expanding the arsenal of treatments available to oncologists [41]. One of the most promising applications of AI in this area is the identification of neoantigens, which are unique to tumor cells and can be targeted by personalized vaccines [36]. Neoantigens arise from tumor-specific mutations and are not present in normal tissues, making them ideal targets for immunotherapy [65]. However, identifying neoantigens is a complex and time-consuming process, requiring the analysis of large-scale genomic data [36]. AI algorithms have been developed to rapidly analyze a patient’s tumor genome and identify neoantigens with high accuracy [36,66]. For example, AI-driven tools have been used to predict which neoantigens are most likely to be recognized by the patient’s immune system, paving the way for the development of personalized cancer vaccines [66].

AI is also being used to discover new immune checkpoints and other targets that could be exploited for cancer immunotherapy [67]. For instance, DL models have been applied to analyze gene expression data and identify novel immune-regulatory pathways that could be targeted to enhance anti-tumor immunity [68]. These discoveries could lead to the development of new classes of immunotherapeutic agents, expanding the range of options available to clinicians and improving patient outcomes. In addition to identifying new therapeutic targets, AI can also help prioritize these targets for further development [41]. By integrating data from various sources, including clinical trials, preclinical studies, and real-world evidence, AI models can assess the potential impact of targeting specific pathways or molecules, thereby guiding the development of new drugs [69]. This ability to prioritize targets based on their predicted clinical relevance could accelerate the translation of research findings into effective therapies, ultimately benefiting patients.

## 4. AI Metrics and Comparative Analysis in Cancer Immunotherapy

To comprehensively evaluate the impact of AI in cancer immunotherapy, it is essential to consider specific performance metrics such as accuracy, area under the curve (AUC), precision, recall, sensitivity, and specificity [70]. These metrics provide a quantitative basis for assessing the effectiveness of AI models and their ability to outperform traditional, non-AI approaches in critical applications [70]. The inclusion of these metrics highlights the tangible advancements AI has brought to cancer treatment, including increased precision, speed, and scalability [70].

### 4.1. Biomarker Identification

AI models, particularly those utilizing ML and DL, have achieved accuracy rates of 85% to 95% in identifying predictive biomarkers for immunotherapy, such as PD-L1 expression, TMB, and MSI [71,72]. For example, using Random Forest models, studies have demonstrated improved stratification of patients likely to benefit from ICIs, directly impacting personalized treatment strategies [73] (see Table 1). Also, DL models such as Convolutional Neural Networks (CNNs) and Recurrent Neural Networks (RNNs) are also employed to process and analyze large-scale datasets [73]. CNNs are particularly effective for spatial and structural data, such as imaging-based biomarker detection, while RNNs are used for sequential data, capturing temporal dynamics [73]. These methods enable the discovery of complex biomarkers, such as gene expression signatures, which are predictive of ICI responses [73] (see Table 1).

Additionally, the AUC values for these models range between 0.70 and 0.95, reflecting high reliability in distinguishing responders from non-responders to treatments like ICIs [71]. In contrast, traditional biomarker discovery methods, which rely heavily on manual analysis and conventional statistical approaches, often yield lower accuracy, are time-intensive, and lack reproducibility [71]. The ability of AI to integrate multi-omics data has revolutionized biomarker identification by uncovering complex patterns not apparent through traditional methods [74].

### 4.2. Patient Response Prediction

AI models, such as those using supervised learning algorithms like Random Forest and Support Vector Machines (SVM), demonstrate significant advantages in predicting patient responses to immunotherapy [75]. Gradient Boosting Machines, Logistic Regression models, and k-Nearest Neighbors (k-NN) are widely used ML techniques for analyzing diverse patient-specific data, including immune signatures, imaging data, and genomic profiles [76]. These models provide robust predictions of patient responses to therapies such as PD-1/PD-L1 inhibitors [76]. Also, DL models, such as Multi-Layer Perceptrons (MLPs), further enhance prediction accuracy by capturing complex, nonlinear interactions in high-dimensional datasets [77]. For example, MLPs have been shown to predict treatment outcomes with high precision, allowing clinicians to avoid ineffective treatments (see Table 1).

These models achieve an AUC mean value of 0.82, enabling precise stratification of patients likely to benefit from ICIs or other therapies [76]. Traditional non-AI methods, such as static predictive models or reliance on clinical judgment, lack this level of granularity and often fail to account for the dynamic and multifactorial nature of tumor-immune interactions [78]. The use of AI for patient response prediction reduces the risk of ineffective treatments and minimizes unnecessary side effects, improving overall therapeutic outcomes [79].

### 4.3. Therapeutic Target Discovery

In the discovery of novel therapeutic targets, DL models using multi-layer neural networks achieve precision rates up to 98.58% and a significant decrease in recall rates, surpassing traditional laboratory-based methods [78,80]. AI can rapidly analyze vast datasets, including genomic and epigenetic information, to identify actionable targets such as neoantigens or novel immune checkpoints [48]. Unsupervised ML techniques such as Clustering and Principal Component Analysis (PCA) are used to identify underlying structures in datasets [81]. These methods group similar data points, enabling the identification of unique tumor profiles and actionable therapeutic targets [81]. Advanced DL models, particularly Generative Adversarial Networks (GANs), are employed to simulate tumor microenvironments and identify neoantigens—tumor-specific proteins that can be targeted by personalized cancer vaccines [82]. GANs are uniquely capable of generating realistic data distributions, which are invaluable for modeling rare or underrepresented tumor types [82]. For example, GANs have been utilized to identify novel immune checkpoints, as described under Discovering Novel Therapeutic Targets in Table 1. Traditional approaches to target discovery are often limited by the scale and complexity of data, leading to slower identification rates and fewer actionable discoveries [43]. AI accelerates this process, providing clinicians and researchers with a broader arsenal of therapeutic options [43].

### 4.4. Drug Development

In drug discovery, AI models such as generative adversarial networks [GANs] and virtual screening techniques have shown remarkable improvements [75]. These models show more efficiency than traditional screening methods [75]. Drug discovery has been transformed by AI through techniques like Graph Neural Networks (GNNs) and virtual screening algorithms [83]. These DL models simulate molecular interactions, enabling rapid identification of candidate drugs for cancer immunotherapy. For instance, GNNs are used to predict drug-target-binding affinities, which significantly accelerates the discovery of novel checkpoint inhibitors and monoclonal antibodies. Virtual screening powered by these models has been shown to achieve a higher hit rate compared to traditional methods [83,84] [see Table 1]. AI reduces the time and cost associated with developing new immunotherapeutic agents by simulating molecular interactions and optimizing drug candidates, enabling faster transitions from research to clinical trials [85].

Furthermore, ML has significantly advanced molecular docking analyses, revolutionizing the drug development process for immunotherapy applications [86]. Docking analyses involve predicting the interaction between molecules, such as how a drug binds to a target protein, which is critical for designing effective immunotherapeutic agents [86]. Traditional docking methods often face challenges in predicting accurate 3D structures due to the complexity of molecular interactions [86]. However, ML tools, including deep neural networks and reinforcement learning models, have substantially improved the reliability and precision of these predictions [87]. By analyzing vast datasets of protein–ligand interactions, ML algorithms can accurately model 3D binding conformations, even for complex immunotherapy targets such as immune checkpoints and neoantigens [86]. These advancements not only enhance the accuracy of docking analyses but also accelerate the identification of high-affinity molecules, reducing the time and costs associated with traditional drug discovery methods [86]. The ability of ML to streamline and improve docking predictions is crucial in developing next-generation immunotherapeutic agents, such as immune checkpoint inhibitors and personalized cancer vaccines, thereby transforming the landscape of drug development in immunotherapy [86].

### 4.5. Clinical Trial Optimization

Optimizing clinical trials, particularly for patient recruitment, relies on ML techniques such as Decision Trees and Random Forests, as well as natural language processing (NLP) models like Transformers [88]. These methods analyze EHRs, genetic profiles, and demographic data to identify eligible candidates [88]. DL algorithms such as Transformer-based models also enable the automated analysis of large-scale clinical data, improving trial design and execution [88]. This is particularly critical for rare cancers, where patient populations are small and difficult to stratify [88]. As noted in Table 1, these techniques enhance recruitment precision and improve trial success rates.

Predictive models used in this context achieve an exclusion accuracy of 95.7% and a recruitment precision of 91.6%, addressing common challenges in enrolling suitable candidates, especially for rare cancers [88]. Traditional trial designs often struggle to meet recruitment targets and adequately stratify patients based on complex criteria [88]. AI not only enhances recruitment efficiency but also enables dynamic adjustments to trial protocols, improving trial success rates and reducing costs [88].

### 4.6. Real-Time Monitoring

In real-time monitoring, AI systems integrating IoT-enabled wearables and time-series analysis deliver high accurate and specificity rates [89]. These metrics reflect the capability of AI to detect physiological changes and predict adverse reactions in real time, allowing for timely interventions [89]. Real-time monitoring of patients undergoing immunotherapy leverages time-series analysis models and DL techniques like Long Short-Term Memory (LSTM) networks. These models process continuous data streams from wearable devices, imaging, and EHRs to detect early signs of adverse reactions or treatment responses [89]. For example, LSTMs are used to predict irAEs based on dynamic physiological data, allowing for timely interventions [89] (see Table 1). Non-AI monitoring systems, which rely on periodic assessments, often fail to provide predictive insights, delaying responses to critical events [89].

## 5. Future Directions

The application of AI in cancer immunotherapy is poised for significant advancements, promising to further refine and enhance the effectiveness of cancer treatments [30]. As AI technology continues to evolve, several promising avenues for research and development are emerging, each with the potential to transform the landscape of cancer care [72,90]. Table 2 provides a detailed exploration of the future directions for AI in cancer immunotherapy, outlining the specific benefits, ongoing research efforts, and challenges associated with each direction.

### 5.1. Real-Time Data Analysis

One of the most exciting prospects for the future of AI in cancer immunotherapy is the integration of real-time data analysis [91,94]. Currently, patient monitoring during immunotherapy is often periodic, relying on scheduled clinical visits and routine tests. However, continuous monitoring of patients could provide a more dynamic and responsive approach to treatment. AI can play a pivotal role in this by analyzing real-time data from various sources such as wearable devices, EHRs, and mobile health applications [91]. Wearable devices can monitor a range of physiological parameters, including heart rate, temperature, and activity levels, which can provide early indicators of adverse reactions to immunotherapy or signals that a treatment is working effectively. By feeding this data into AI algorithms, clinicians can receive alerts about potential issues before they become critical, allowing for timely interventions. For example, if a patient shows early signs of an immune-related adverse event, an AI system could recommend a change in medication or dosage, potentially preventing more severe complications [30].

Moreover, the integration of real-time data analysis could enable more precise adjustments to treatment regimens [91]. For instance, AI could analyze fluctuations in biomarkers or tumor response as they happen, allowing oncologists to adjust the type or intensity of immunotherapy in real time, thereby optimizing treatment efficacy and reducing unnecessary side effects [41]. This real-time approach could also facilitate adaptive clinical trials, where AI continuously assesses data to refine treatment protocols as the trial progresses, increasing the likelihood of success and accelerating the development of new therapies [91].

### 5.2. AI-Driven Clinical Trials

Clinical trials are a critical component of developing new cancer therapies, but they are often time-consuming, expensive, and logistically challenging. AI has the potential to revolutionize the design and execution of clinical trials, making them more efficient and effective. One of the primary ways AI can impact clinical trials is through the optimization of patient recruitment [40,92,96]. Currently, patient recruitment is a significant bottleneck in clinical trials, with many trials failing to meet enrollment targets [92]. AI can analyze large datasets, including EHRs, genetic information, and demographic data, to identify suitable candidates for specific trials more quickly and accurately [40]. This targeted approach can ensure that trials are populated with patients who are most likely to benefit from the treatment being studied, thereby improving the quality and relevance of trial outcomes [96].

AI can also optimize trial design by simulating various scenarios and predicting the most effective trial protocols. For example, AI models can analyze historical trial data to identify patterns and factors that contributed to the success or failure of previous studies [40,97]. These insights can then be used to design trials that are more likely to succeed, reducing the need for multiple phases of testing and shortening the time required to bring new treatments to market [40]. Additionally, AI can be used to analyze interim trial data in real time, allowing for dynamic adjustments to trial protocols based on ongoing results. This adaptability can enhance the efficiency of trials and reduce costs, ultimately speeding up the development of new immunotherapies. Furthermore, AI-driven analysis of clinical trial data can improve the accuracy and granularity of trial outcomes [98]. Traditional methods of analyzing trial data often rely on predefined endpoints and statistical methods that may not fully capture the complexity of patient responses. AI can analyze vast amounts of trial data, identifying subtle patterns and correlations that might be missed by conventional analysis. This can lead to a deeper understanding of how and why certain treatments work, providing valuable insights that can inform future research and clinical practice [99].

### 5.3. Personalized Immunotherapy

The ultimate goal of AI in cancer immunotherapy is to achieve fully personalized treatment plans tailored to the unique characteristics of each patient’s cancer. Personalized immunotherapy aims to maximize efficacy while minimizing side effects by accounting for the individual genetic, molecular, and clinical features of the patient’s tumor and immune system [54]. AI is uniquely suited to this task because of its ability to integrate and analyze complex datasets from multiple sources. AI can combine data from genomic sequencing, transcriptomic analysis, proteomic profiling, and clinical records to create a comprehensive profile of each patient’s cancer [68]. This profile can then be used to predict how the patient will respond to different immunotherapies, allowing clinicians to choose the most effective treatment regimen. For example, AI can identify specific mutations or neoantigens that make a tumor more susceptible to certain ICIs, or it can suggest combinations of therapies that are likely to produce a synergistic effect [36].

AI serves as a robust tool for detecting tumors with mismatch repair (MMR) deficiency from histopathology slides, significantly advancing pathological evaluation and patient eligibility for immunotherapy [47]. Through the analysis of digital slides, AI models can predict key pathological features, such as Gleason scores in prostate cancer patients, which directly inform treatment decisions [47]. This capability is critical for identifying patients who are likely to benefit from ICI and other immunotherapies [100].

Recent studies have further highlighted the potential of AI in PD-L1 expression and tumor-infiltrating lymphocyte (TIL) assessments [47,100]. PD-L1 expression levels, determined through immunohistochemistry, are a key biomarker for ICI eligibility. AI models have demonstrated exceptional accuracy in automating the evaluation of PD-L1 levels, reducing interobserver variability, and enhancing reproducibility in clinical practice [100]. Similarly, AI-based methods for TIL quantification provide insights into the immune microenvironment, which is crucial for determining tumor response to immunotherapy. These tools standardize TIL counting, offering a more objective approach compared to manual evaluations.

Additionally, AI tools can predict MMR status or susceptibility to ICI therapy by analyzing morphological patterns in hematoxylin and eosin (H&E)-stained slides. Studies have shown that AI can identify subtle histological features associated with MMR deficiency, such as nuclear atypia and stromal changes, that may not be apparent through traditional pathological assessments [100]. These models have achieved promising results in predicting MMR status with high sensitivity and specificity, thereby streamlining patient selection for immunotherapy.

By integrating these capabilities, AI not only enhances diagnostic precision but also expedites the identification of eligible patients for advanced cancer treatments. Its application in pathological evaluation represents a significant step toward more personalized and effective immunotherapy strategies, ensuring that patients receive the most suitable and timely interventions.

In addition to selecting the appropriate therapy, AI can also personalize the dosing and scheduling of treatments [43]. Traditional dosing regimens are often based on population averages, which may not be optimal for every patient [43]. AI can analyze real-time data on a patient’s response to therapy, adjusting the dosage or timing of treatments to maximize effectiveness and minimize toxicity [43,101]. This approach, known as adaptive dosing, could significantly improve patient outcomes, particularly for those with complex or advanced cancers. Moreover, personalized immunotherapy could extend beyond treatment selection to include the development of custom therapies [36]. For instance, AI can aid in the design of personalized cancer vaccines by identifying neoantigens unique to a patient’s tumor [36]. These vaccines can then be tailored to stimulate the patient’s immune system to target and destroy cancer cells more effectively [36]. This level of personalization represents a significant leap forward in the ability to treat cancer, moving from a one-size-fits-all approach to one that is precisely tailored to the individual’s needs.

### 5.4. Benchmarks for Measuring Success in AI-Driven Cancer Immunotherapy

To ensure the effectiveness and practical integration of AI in cancer immunotherapy, benchmarks for success must be clearly defined. These benchmarks allow for systematic evaluation of the progress in addressing key challenges, including personalized immunotherapy, drug discovery, and patient monitoring.

For biomarker identification, success can be measured by the precision and recall of AI models in predicting biomarkers such as PD-1, TMB, and MSI, compared to traditional methods [33]. Metrics like the AUC and F1 score provide quantifiable benchmarks, with clinical trials validating whether AI-identified biomarkers lead to improved patient stratification and treatment outcomes [33].

In personalized immunotherapy, the effectiveness of AI-generated treatment plans can be assessed through clinical trials comparing progression-free survival and overall survival rates of patients receiving AI-guided treatments versus conventional approaches [75]. The success of these plans can be determined by statistically significant improvements in patient outcomes and their adoption in real-world clinical settings [75].

AI-driven drug discovery benchmarks include the time and cost reductions achieved in identifying and validating new drug candidates [102]. The success of AI in this domain can be measured by reductions in the average drug discovery timeline and costs, as well as the transition of AI-discovered candidates to the Food and Drug Administration (FDA) or the European Medicines Agency (EMA) approvals after demonstrating safety and efficacy in Phase I clinical trials [102]. Similarly, monitoring treatment response involves assessing the real-time accuracy of AI in detecting irAEs [102]. A reduction in false positive and negative rates and improvements in patient safety metrics, such as fewer hospitalizations due to early adverse event detection, would indicate success [102].

For clinical trial optimization, the efficiency of patient recruitment and stratification using AI can be measured by increased enrollment rates, reduced recruitment timelines, and enhanced trial success due to precise matching of patients to suitable trials [103]. Faster trial completion and successful regulatory milestones validate AI’s contribution to this area [103].

While the focus of this review is on traditional AI methods, generative AI’s potential role in personalized immunotherapy can be evaluated by comparing the efficacy of AI-generated treatment plans through randomized controlled trials, ensuring regulatory compliance, and gauging clinician adoption rates. Clear benchmarks like these enable systematic evaluation of AI’s progress and its impact on cancer immunotherapy, ensuring its ethical and practical integration into clinical workflows.

## 6. Limitations and Challenges

Despite its transformative potential, integrating AI into cancer immunotherapy is fraught with several limitations and challenges that must be addressed to unlock its full benefits. These limitations span technical, regulatory, logistical, and ethical dimensions, each requiring targeted solutions.

One of the foremost limitations is data availability, quality, and diversity. AI models are inherently data-driven, relying on large, high-quality datasets for training and validation. However, access to such data is restricted by privacy regulations, institutional barriers, and ethical concerns [104]. Even when data are available, they often suffer from variability in formats, incomplete records, and imbalances in demographic representation, particularly for underrepresented populations and rare cancer types [104]. This lack of diversity can lead to biased AI predictions, reducing the generalizability and fairness of AI applications across different patient groups.

Another significant challenge is algorithm transparency and interpretability. Many advanced AI systems, particularly deep-learning models, operate as “black boxes”, providing accurate predictions but lacking clear explanations for their outputs [105]. This opacity is a critical barrier in clinical settings, where clinicians need to trust and understand AI-driven recommendations [105]. Without interpretability, the clinical adoption of AI tools becomes challenging, as medical professionals are hesitant to rely on systems they cannot fully explain to patients or validate within their existing frameworks [105].

Regulatory and validation challenges further complicate the pathway for AI integration. Regulatory agencies such as the FDA and EMA require rigorous evidence to approve AI tools as Software as a Medical Device (SaMD) [105]. This involves compliance with stringent standards like ISO 13485 for quality management and IEC 62304 for software life cycle processes, alongside robust prospective clinical trials or multi-center studies [106]. These processes are resource-intensive, expensive, and time-consuming, leading to a bottleneck in bringing AI-driven innovations to clinical practice [105]. Additionally, regulatory frameworks for AI in healthcare are still evolving, creating uncertainty for developers and delaying widespread adoption [105].

The logistical challenges of clinical implementation also represent a significant hurdle [106]. For AI systems to be effectively deployed, healthcare infrastructures must support standardized data formats and seamless interoperability between AI platforms and existing EHR systems [106]. Moreover, healthcare professionals need adequate training to use these tools effectively, understand their outputs, and integrate them into clinical workflows [106]. Many institutions lack the resources or expertise to address these requirements, slowing the adoption process [106].

From an ethical and legal standpoint, concerns about data privacy, patient consent, and the potential misuse of AI models remain paramount. AI systems require sensitive patient data for training, raising questions about how this data is stored, shared, and protected [104]. Without clear ethical guidelines and legal safeguards, breaches of data privacy could lead to loss of public trust and legal repercussions [104]. Furthermore, AI’s ability to automate aspects of healthcare delivery raises concerns about accountability—particularly if an AI-driven decision results in an adverse outcome [104].

Finally, technical challenges in AI model development must be acknowledged. AI systems require continuous updates and retraining as new data become available, necessitating robust feedback loops and monitoring mechanisms [105]. Moreover, handling real-world data, which is often noisy and incomplete, poses additional difficulties in ensuring the reliability and reproducibility of AI models [105].

Addressing these limitations will require a multidisciplinary approach involving collaboration among technologists, clinicians, regulators, and policymakers. Efforts must focus on enhancing data accessibility while ensuring privacy, developing explainable AI models, establishing clear regulatory pathways, and creating robust infrastructures to integrate AI into clinical workflows. By addressing these challenges systematically, the promise of AI in revolutionizing cancer immunotherapy can be realized safely and equitably.

## 7. Conclusions

AI is transforming cancer immunotherapy by enhancing treatment efficacy, predicting patient responses, and discovering novel therapeutic targets. Its integration into oncology is driving a shift toward personalized and precise care. However, realizing AI’s potential depends on addressing key challenges, including ensuring data privacy, algorithm transparency, and clinician trust. Regulatory compliance is crucial, with few AI applications currently achieving FDA clearance or CE marking. Validation through prospective clinical trials and adherence to rigorous standards, such as ISO 13485 and IEC 62304, is essential for safe implementation in clinical workflows.

As logistical barriers such as data standardization, system interoperability, and clinician training are resolved, AI is poised to play a central role in cancer care. By personalizing treatments, optimizing clinical trials, and providing real-time insights, AI offers the potential for more effective therapies and improved patient outcomes. While challenges remain, the progress being made underscores the promise of AI as a transformative force in oncology, paving the way for safer and more impactful integration into clinical practice.

## Figures and Tables

**Figure 1 diseases-13-00024-f001:**
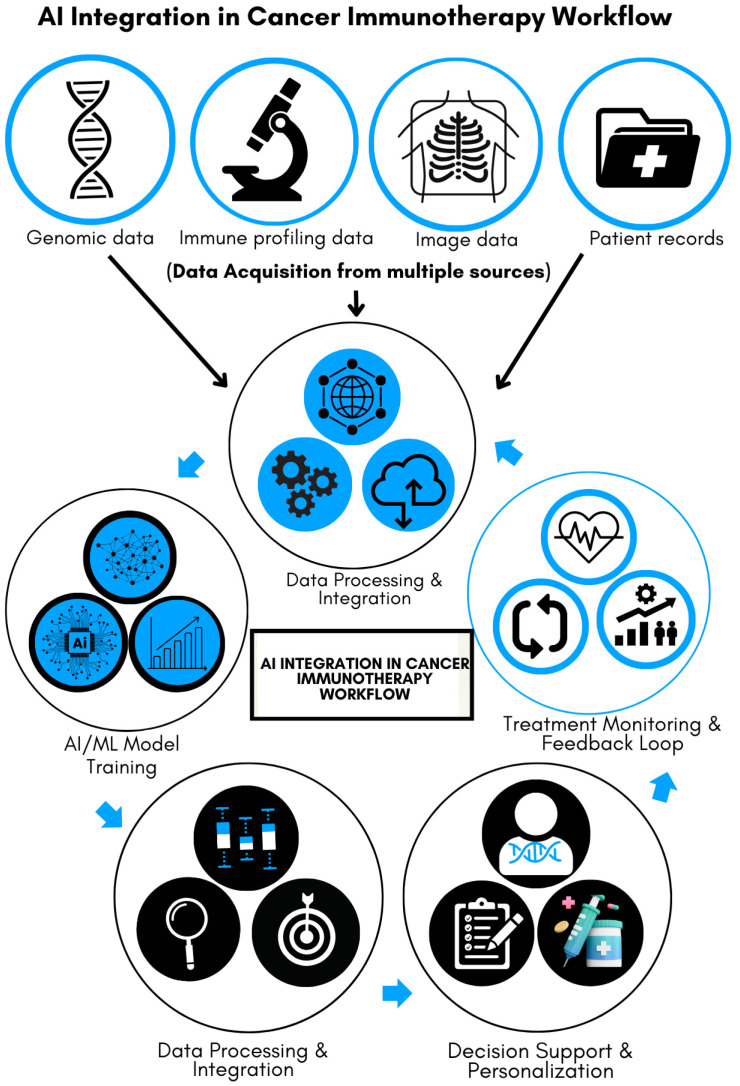
AI integration in cancer immunotherapy workflow.

**Figure 2 diseases-13-00024-f002:**
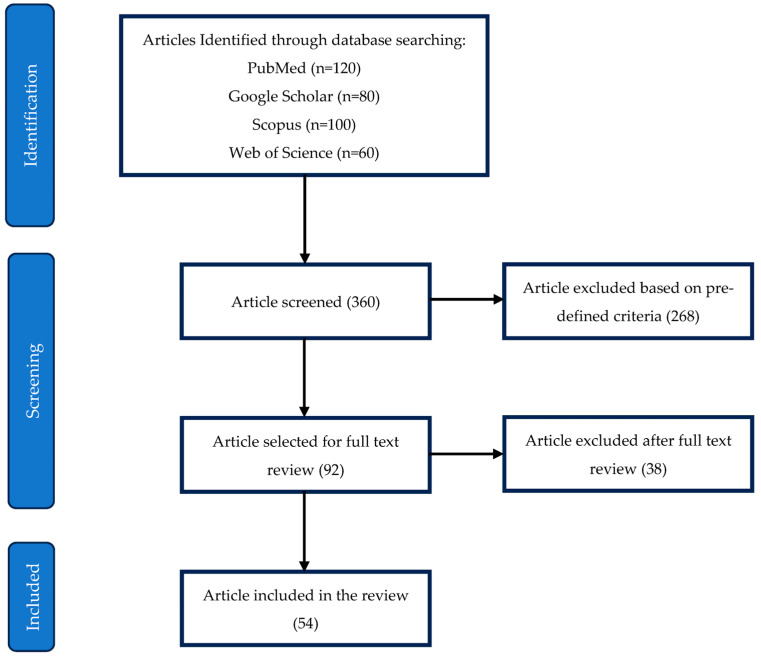
PRISMA flowchart for selection of included studies.

**Figure 3 diseases-13-00024-f003:**
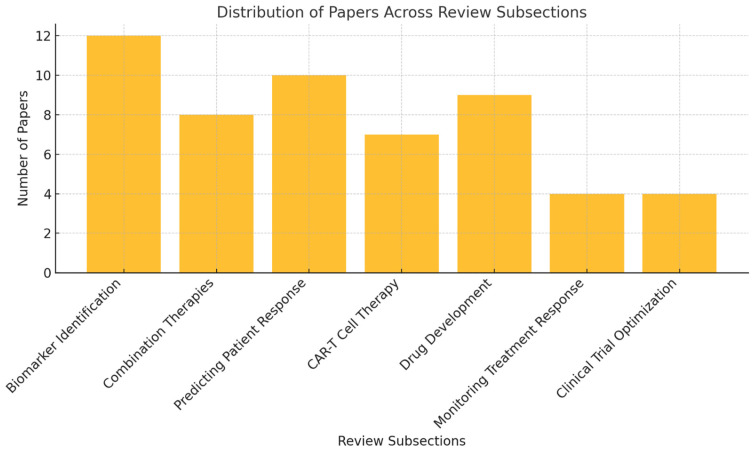
Distribution of papers across key subsections of the review.

**Figure 4 diseases-13-00024-f004:**
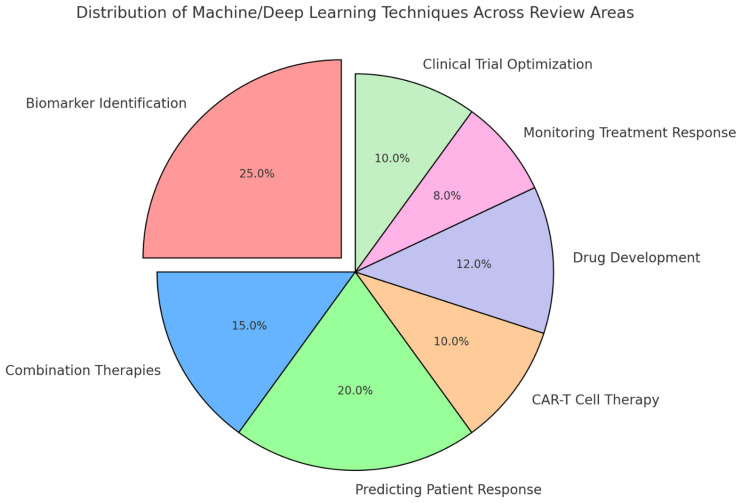
Distribution of machine- and deep-learning techniques across review areas.

**Figure 5 diseases-13-00024-f005:**
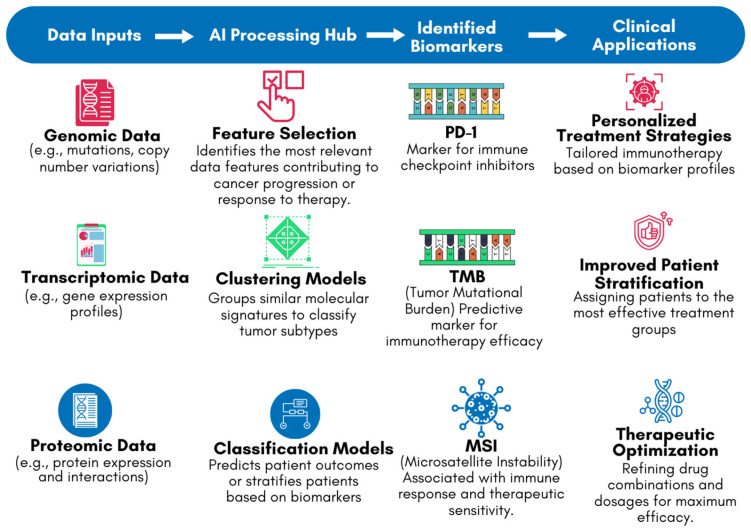
AI-driven biomarker discovery workflow for cancer immunotherapy.

**Table 1 diseases-13-00024-t001:** Comprehensive overview of AI applications in cancer immunotherapy.

Application	Description	Methodology	Examples	Impact on Treatment
Biomarker Identification [30]	AI analyzes complex biological data to discover biomarkers that predict immunotherapy response	Supervised ML algorithms (e.g., Random Forest, Support Vector Machines) and DL algorithms (e.g., Convolutional Neural Networks, Recurrent Neural Networks) applied to multi-omics data [genomic, transcriptomic, proteomic)	Identification of TMB, MSI, PD-L1 expression levels, and gene signatures predictive of ICI response	Enables personalized treatment by stratifying patients based on likely treatment response, improving efficacy, and minimizing adverse effects
Optimization of Combination Therapies [33]	AI predicts the most effective combinations of immunotherapy with other treatments [e.g., chemotherapy, radiation]	Reinforcement learning models and Bayesian networks applied to clinical trial data, real-world evidence, and patient-specific data	Successful combinations of ICIs with chemotherapy or radiation therapy in lung cancer and melanoma	Reduces reliance on trial and error, accelerates the identification of optimal treatment regimens, and enhances overall therapeutic outcomes
Predicting Patient Response [30,34,35]	AI models forecast which patients will benefit from specific immunotherapy treatments and who might experience severe side effects	Gradient Boosting Machines, Logistic Regression, and DL models (e.g., Multi-Layer Perceptrons) using patient data (e.g., genomic profiles, immune signatures, imaging data)	ML predictions for PD-1/PD-L1 inhibitor responses in melanoma and non-small cell lung cancer	Guides clinical decision-making, allowing for more precise, personalized treatment, and avoiding unnecessary side effects
Discovering Novel Therapeutic Targets [30,36]	AI uncovers new targets for immunotherapy by analyzing vast biological datasets	Unsupervised ML models (e.g., Clustering, Principal Component Analysis) and Generative Adversarial Networks (GANs) applied to genetic, epigenetic, and transcriptomic data	Identification of neoantigens and novel ICIs for developing personalized vaccines and new therapeutic agents	Expands the range of therapeutic options, leading to the development of new immunotherapy agents tailored to specific tumor types
Enhancing Drug Discovery [37,38]	AI accelerates the discovery and development of new immunotherapeutic agents	Virtual screening, DL molecular modeling (e.g., Graph Neural Networks), and simulation of drug interactions	AI-driven discovery of novel checkpoint inhibitors and monoclonal antibodies for cancer treatment	Shortens the drug development timeline, reducing costs and bringing effective treatments to market more quickly
Monitoring Treatment Response [39]	AI facilitates real-time monitoring of patient responses during immunotherapy	Time-series analysis models and DL algorithms (e.g., Long Short-Term Memory networks) applied to data from wearables, imaging, and EHR	AI-integrated wearable devices monitor physiological changes and detect early signs of adverse reactions	Enables timely interventions, optimizing treatment outcomes and improving patient safety during therapy
Improving Patient Selection for Clinical Trials [40]	AI identifies suitable candidates for clinical trials based on predictive models	Natural Language Processing (NLP), supervised ML models (e.g., Decision Trees), and DL models (e.g., Transformers) applied to EHR, genetic profiles, and clinical history	Enhanced recruitment for immunotherapy trials, particularly for rare cancer types or specific genetic subtypes	Increases the efficiency and success rate of clinical trials, ensuring that trials are populated with the most suitable candidates
Adaptive Treatment Strategies [41]	AI supports dynamic adjustment of treatment plans based on ongoing patient data	Adaptive reinforcement learning and real-time DL models integrating data from multiple sources for adjusting treatment parameters	AI-driven adaptive dosing and sequencing strategies in immunotherapy to enhance effectiveness and reduce toxicity	Personalizes treatment regimens, improving outcomes by adapting to individual patient responses over time

Abbreviations: Artificial Intelligence, AI; Tumor Mutational Burden, TMB; Machine Learning, ML; Deep Learning, DL; Microsatellite Instability, MSI; Programmed Cell Death Ligand 1, PD-L1; Immune Checkpoint Inhibitors, ICIs; Programmed Cell Death Protein 1, PD-1; Electronic Health Records, EHR.

**Table 2 diseases-13-00024-t002:** Comprehensive future directions for AI in cancer immunotherapy.

Future Direction	Description	Potential Benefits	Challenges to Address
Adaptive Treatment Strategies [30]	AI enables dynamic adjustment of treatment protocols based on real-time patient response data, ensuring optimal dosing and timing	Improves patient outcomes by continuously adapting treatment plans to changing patient conditions, reducing toxicity, and enhancing efficacy	Developing robust real-time monitoring systems, managing the computational demands of real-time data processing, and ensuring clinical acceptance of AI-guided adaptive protocols
Personalized Immunotherapy [30,68]	AI creates tailored treatment plans by integrating multi-omics data (genomic, transcriptomic, proteomic) with clinical data for each patient	Maximizes treatment efficacy and minimizes side effects by delivering personalized treatment regimens	Complexity in integrating diverse data sources, ensuring the interpretability of AI-generated treatment plans, and gaining regulatory approval for personalized approaches
AI-Driven Clinical Trials [40,91,92]	Use of AI to optimize all stages of clinical trials, from patient recruitment and trial design to data analysis and outcome prediction	Reduces the time and cost associated with clinical trials, improves patient matching, and enhances the likelihood of trial success	Standardization of trial protocols across different regions, regulatory approval, and ensuring the generalizability of AI-driven trial outcomes across diverse patient populations
AI-Enhanced Imaging for Immunotherapy [54]	AI improves the interpretation of imaging data (e.g., PET scans, CT scans) to assess tumor response to immunotherapy more accurately	Provides more precise evaluations of treatment effectiveness, leading to better-informed clinical decisions and adjustments in therapy	Addressing variability in imaging quality, ensuring the integration of AI-driven imaging with other clinical data, and gaining clinician trust in AI interpretations
AI in Drug Discovery and Development [69,93]	AI accelerates the identification and development of new immunotherapeutic agents by predicting molecular interactions and simulating drug responses	Shortens the drug development timeline, reduces costs, and enhances the precision of drug-target interactions, leading to more effective treatments	Addressing the accuracy of AI predictions, regulatory challenges in drug approval, and the need for extensive validation studies to confirm AI-generated drug candidates’ efficacy and safety
Real-Time Data Analysis [91,94]	Integration of AI with real-time data from wearable devices, EHRs, and mobile apps to continuously monitor patients during immunotherapy	Enables continuous, personalized monitoring of patient health, leading to timely interventions and optimized treatment outcomes	Ensuring data privacy and security, managing data overload, and developing algorithms that can accurately interpret and act on real-time data in a clinical setting
AI for Predictive Toxicology [95]	AI predicts potential toxicities and side effects of new immunotherapies before clinical trials, reducing the risk of adverse effects	Enhances patient safety, reduces trial failure rates due to toxicity, and accelerates the development of safer immunotherapeutic drugs	Ensuring the robustness and accuracy of AI models in predicting rare toxicities, integrating these predictions into the drug development pipeline, and balancing predictive sensitivity with clinical relevance
AI in Immune System Modeling	AI models complex immune system interactions to better understand how tumors evade immune detection and how therapies can be optimized	Provides deeper insights into immune–tumor interactions, leading to the development of more effective immunotherapy strategies	Complexity in modeling the immune system accurately, ensuring that AI models are based on validated biological principles, and integrating these models with clinical decision-making processes

Abbreviations: Artificial Intelligence, AI; Electronic Health Records, EHR; Positron Emission Tomography, PET; Computed Tomography, CT.

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
