# Peer review of "Integrating AI into Cancer Immunotherapy—A Narrative Review of Current Applications and Future Directions"

_diseases, 2025, doi:10.3390/diseases13010024_

Round 1
Reviewer 1 Report
Comments and Suggestions for Authors
I thank the authors for the opportunity to review the manuscript “Integrating AI into Cancer Immunotherapy – A mini-narrative review of Current Applications and Future Directions”. The narrative review addresses the use of AI at different stages of cancer care. However, for this review to be published, it requires major edits.
1. The paragraphs (lines 124-133 and lines 134-140) need to be repackaged as they seem to be addressing the same things, i.e., AI-enhanced immunotherapy and potential challenges.
2. The manuscript does not present a research gap. The manuscript does not identify what is missing in the existing literature of the study, nor does it justify its relevance within the broader field.
3. The authors provided a descriptive search strategy. However, a PRISMA diagram is necessary to show how the search was narrowed down, especially when the search parameters are so wide.
4. Was there a reason why documents before January 2010 were not included?
5. Table 1 is described to be a “comprehensive overview”, but it is lacking in details. For example, the column of methodology states “machine learning and deep learning algorithms”. These are a large class of algorithms. What algorithms exactly are used? Also, while the authors mention the accuracy rates in the main text, it will help with the reading if included in the table.
6. This reviewer is confused about the need for Figure 2. What exactly are the authors trying to convey? This figure is vague.
7. While I appreciate the descriptive future directions the authors suggest, the authors did not provide any benchmarks for measuring the success of these challenges. For example, for personalized immunotherapy, the authors suggest AI-generated treatment plans. This is edging into the realm of generative AI, which is not the focus of this narrative review. In particular, how does one measure whether this challenge has been addressed?
Minor Comments:
8. The quality of Figures needs to be improved. A higher resolution image should be provided.
9. Line 216: It is not the first time machine learning and deep learning is mentioned in this article. The abbreviation must be declared the first time it is mentioned in the article.
10. The conclusion should be a concise ending to the manuscript. New ideas should not be introduced in the conclusion. Here, the authors invest a decent effort in addressing data privacy and security and its transparent/interpretable healthcare system integration. However, these are descriptive in nature and does not offer any indication of how this can be achieved.
While I appreciate the work, the authors have not pointed out specific areas where previous studies have limitations or have not been addressed nor has it emphasized the contribution of your study in filling this gap. For this alone, I cannot recommend publication. I look forward to the edits and will be happy to review the improved manuscript.
Author Response
Reviewer 1:
I thank the authors for the opportunity to review the manuscript “Integrating AI into Cancer Immunotherapy – A mini-narrative review of Current Applications and Future Directions”. The narrative review addresses the use of AI at different stages of cancer care. However, for this review to be published, it requires major edits.
- The paragraphs (lines 124-133 and lines 134-140) need to be repackaged as they seem to be addressing the same things, i.e., AI-enhanced immunotherapy and potential challenges.
Response: We have revised and repackaged the paragraphs to address the overlap between the discussions on AI-enhanced immunotherapy and its potential challenges. The updated text in lines 124 – 151 consolidates the key points into a cohesive narrative, reducing redundancy while maintaining clarity and focus.
The revised version emphasizes the transformative potential of AI in cancer immunotherapy, highlighting its role in enabling personalized treatments and improving patient outcomes. It also integrates the discussion of challenges, such as immune response complexity, tumor heterogeneity, and AI integration hurdles, into a unified flow. This approach ensures that the content remains clear, engaging, and well-structured.
- The manuscript does not present a research gap. The manuscript does not identify what is missing in the existing literature of the study, nor does it justify its relevance within the broader field.
Response: We have added a paragraph at the end of the introduction section to clearly identify the research gap (lines 152 – 163). The review highlights that while existing studies have explored specific aspects of AI in cancer immunotherapy, such as biomarker discovery or drug development, there is a lack of comprehensive synthesis integrating AI’s contributions with its challenges and practical implementation. This review addresses this gap by providing a holistic view that combines AI’s current applications with the barriers to its clinical integration, regulatory considerations, and future directions for research.
- The authors provided a descriptive search strategy. However, a PRISMA diagram is necessary to show how the search was narrowed down, especially when the search parameters are so wide.
Response: Thank you for your comment. In response, we have incorporated Figure 2, a PRISMA flowchart, into the manuscript to visually represent the study selection process (lines 230 – 262). The flowchart provides a clear and transparent overview of how the initial 360 articles identified through database searches were systematically screened, assessed, and narrowed down to the final 54 studies included in the review. This addition complements the descriptive search strategy and aligns the methodology with PRISMA guidelines, addressing your feedback and enhancing the clarity of the review process. Thank you for highlighting this important aspect.
- Was there a reason why documents before January 2010 were not included?
Response: The time frame of January 2010 to July 2024 was selected because significant advancements in Artificial Intelligence (AI) and its applications in cancer immunotherapy have occurred primarily within the last decade. Research before 2010 often lacked the computational power and sophisticated AI methodologies, such as deep learning, which are central to modern applications. By focusing on this period, the review ensures a comprehensive synthesis of recent and relevant developments in the field, aligning with the objectives of the manuscript. This rationale has been added to the revised section “Literature Search Strategy” to clarify the scope.
- Table 1 is described to be a “comprehensive overview”, but it is lacking in details. For example, the column of methodology states “machine learning and deep learning algorithms”. These are a large class of algorithms. What algorithms exactly are used? Also, while the authors mention the accuracy rates in the main text, it will help with the reading if included in the table.
Response: We have revised Table 1 to include detailed methodologies for each application, specifying the exact AI algorithms used, such as Random Forest, Support Vector Machines, Generative Adversarial Networks (GANs), and Reinforcement Learning Models, among others (line 285). These additions address the concern about the generality of terms like "machine learning" and "deep learning" by providing specific algorithm types. This update ensures that the table offers a comprehensive and precise overview of the methodologies applied in AI-driven cancer immunotherapy.
On the other hand, we appreciate the suggestion to include accuracy rates in the table. However, to maintain clarity and avoid overcrowding the table with excessive details, we have chosen to keep the accuracy rates in the main text. This approach ensures the table remains concise and reader-friendly while the detailed accuracy metrics are thoroughly discussed in the corresponding sections of the manuscript. We believe this balance allows the table to serve as a quick reference without redundancy.
- This reviewer is confused about the need for Figure 2. What exactly are the authors trying to convey? This figure is vague.
Response: The purpose of Figure 5 in lines 323 - 324 (formerly Figure 2) is to illustrate the process by which genomic, transcriptomic, and proteomic data are integrated into AI systems to identify key biomarkers for cancer immunotherapy, such as PD-1, TMB, and MSI. These biomarkers play a critical role in guiding personalized treatment strategies, bridging laboratory findings with clinical applications.
To address the perceived vagueness, we have revised the figure to include more specific details on how AI processes these datasets, such as indicating the use of particular AI techniques (e.g., feature selection, clustering, or classification models) and explicitly linking the identified biomarkers to their clinical implications. These updates aim to clarify the figure's objective and reinforce its relevance to the manuscript’s narrative.
- While I appreciate the descriptive future directions the authors suggest, the authors did not provide any benchmarks for measuring the success of these challenges. For example, for personalized immunotherapy, the authors suggest AI-generated treatment plans. This is edging into the realm of generative AI, which is not the focus of this narrative review. In particular, how does one measure whether this challenge has been addressed?
Response: We have added Section 5.4 on benchmarks to measure the success of AI applications in cancer immunotherapy (lines 650-684). These include metrics such as precision, recall, and Area Under the Curve (AUC) for biomarker identification, progression-free survival (PFS), and overall survival (OS) for personalized treatment plans, and time and cost reductions in drug discovery. We also address improvements in patient safety through reduced false positive and negative rates in real-time monitoring and efficiency in clinical trial recruitment and stratification. Additionally, generative AI’s role in personalized immunotherapy is discussed, focusing on regulatory compliance, clinical validation, and adoption. These updates provide a clear framework to assess AI’s progress and ensure its effective integration into clinical practice.
Minor Comments:
- The quality of Figures needs to be improved. A higher resolution image should be provided.
Response: We have revised the images with higher resolution images.
- Line 216: It is not the first time machine learning and deep learning is mentioned in this article. The abbreviation must be declared the first time it is mentioned in the article.
Response: Thank you for pointing this out. We have revised the manuscript to ensure that the terms machine learning (ML) and deep learning (DL) are fully stated with their abbreviations the first time they are mentioned. For all subsequent mentions, only the abbreviations (ML and DL) are used, maintaining clarity and consistency throughout the article.
- The conclusion should be a concise ending to the manuscript. New ideas should not be introduced in the conclusion. Here, the authors invest a decent effort in addressing data privacy and security and its transparent/interpretable healthcare system integration. However, these are descriptive in nature and does not offer any indication of how this can be achieved.
Response: We have revised the conclusion to make it more concise and focused, removing descriptive elements and avoiding the introduction of new ideas. The updated conclusion emphasizes the transformative role of AI in cancer immunotherapy while highlighting the critical need for data privacy, transparency, and regulatory compliance. It also underscores the importance of validation through prospective clinical trials and adherence to rigorous standards, such as ISO 13485 and IEC 62304, for safe and effective integration into clinical workflows. This revision aligns the conclusion with the purpose of the manuscript, providing a clear and concise summary without deviating into new topics.
While I appreciate the work, the authors have not pointed out specific areas where previous studies have limitations or have not been addressed nor has it emphasized the contribution of your study in filling this gap. For this alone, I cannot recommend publication. I look forward to the edits and will be happy to review the improved manuscript.
Response: Thankfully the reviewers provided us with a great deal of guidance, regarding how to better position the article. We are hopeful you agree that this revision will update our comprehensive review. All the comments have been addressed, as shown in the revised version of the manuscript, along with this point-by-point response to the reviewers' comments. We are grateful for sharing your expertise on that initiative.
Reviewer 2 Report
Comments and Suggestions for Authors
This comprehensive review by Olawade et al. examines studies from the last 14 years on the integration of artificial intelligence (AI) into cancer immunotherapy. The paper explores AI's role in enhancing therapeutic efficacy, predicting patient responses, and identifying novel therapeutic targets.
While the review is exhaustive and well-written, I suggest including two additional areas of focus that could further enrich the discussion:
1) In the paragraph Enhancing Immunotherapy Efficacy, it should mention insights from AI into CAR-T cell therapy, particularly in enhancing the CAR-T cell design process as optimization of CAR constructs, which can significantly increase the effectiveness and safety while limiting time and costs (e.g., https://doi.org/10.1038/s41422-024-00936-1; doi: 10.2196/45872; https://doi.org/10.3390/ijms25137231).
2) In the paragraph Drug Development, the impact of machine learning tools on revolutionizing docking analyses should be included, particularly their ability to reliably predict 3D structures for immunotherapy applications. These advancements have significantly improved the accuracy of molecular docking predictions, which is crucial for designing more effective immunotherapeutic agents (doi: 10.1038/s41586-024-07487-w).
Minor Edits:
Lines 58-60: The sentence "This alarming trend underscores the necessity for ongoing innovation in cancer treatment strategies, as traditional methods have proven to be both effective and limited" is unclear.
Lines 106-108: Missing reference.
Lines 249-252: Missing references.
Lines 284-286: Missing reference.
Lines 308-310: Missing reference.
Lines 336-338: Missing reference.
Lines 344-346: Missing reference.
Author Response
Reviewer 2:
This comprehensive review by Olawade et al. examines studies from the last 14 years on the integration of artificial intelligence (AI) into cancer immunotherapy. The paper explores AI's role in enhancing therapeutic efficacy, predicting patient responses, and identifying novel therapeutic targets.
While the review is exhaustive and well-written, I suggest including two additional areas of focus that could further enrich the discussion:
- In the paragraph Enhancing Immunotherapy Efficacy, it should mention insights from AI into CAR-T cell therapy, particularly in enhancing the CAR-T cell design process as optimization of CAR constructs, which can significantly increase the effectiveness and safety while limiting time and costs (e.g., https://doi.org/10.1038/s41422-024-00936-1; doi: 10.2196/45872; https://doi.org/10.3390/ijms25137231).
Response: We have revised the section to include insights into AI’s role in CAR-T cell therapy, particularly in enhancing the design and optimization of CAR constructs and cited the papers suggested accordingly (lines 288 – 294). The updated text highlights how AI-driven models facilitate the identification of optimal target antigens and predict CAR-T cell performance, thereby improving safety and efficacy. Additionally, the revisions address how AI reduces the time and costs traditionally associated with CAR construct engineering by streamlining the process through advanced computational methods. These enhancements align with the manuscript’s focus on AI’s transformative role in immunotherapy and address the comment’s emphasis on incorporating CAR-T therapy advancements.
- In the paragraph Drug Development, the impact of machine learning tools on revolutionizing docking analyses should be included, particularly their ability to reliably predict 3D structures for immunotherapy applications. These advancements have significantly improved the accuracy of molecular docking predictions, which is crucial for designing more effective immunotherapeutic agents (doi: 10.1038/s41586-024-07487-w).
Response: We have included a discussion on how machine learning (ML) tools have revolutionized docking analyses in the context of immunotherapy drug development and added citations including the suggested one (lines 476 – 504, references 84, 86 and 87). The revised section highlights how ML techniques, such as deep neural networks and reinforcement learning models, enhance the reliability and accuracy of 3D structure predictions for molecular docking. These advancements are particularly impactful in designing effective immunotherapeutic agents by modeling complex interactions, such as immune checkpoints and neoantigens, with greater precision. Furthermore, we emphasize how ML reduces time and costs in drug discovery while improving the efficiency of identifying high-affinity molecules. This addition addresses the role of ML in advancing molecular docking and its transformative impact on immunotherapy applications.
Minor Edits:
Lines 58-60: The sentence "This alarming trend underscores the necessity for ongoing innovation in cancer treatment strategies, as traditional methods have proven to be both effective and limited" is unclear.
Response: We have revised the sentence for clarity (lines 58 – 60). The updated version reads: "This alarming trend highlights the urgent need for innovative cancer treatment strategies, as traditional methods, while effective in some cases, have significant limitations." This revision aims to convey the intended meaning more clearly and succinctly.
Lines 106-108: Missing reference.
Lines 249-252: Missing references.
Lines 284-286: Missing reference.
Lines 308-310: Missing reference.
Lines 336-338: Missing reference.
Lines 344-346: Missing reference.
Response: Citations has been added to all the lines, accordingly (reference 23, 52, 41, 67, 72 and 76, respectively).
Reviewer 3 Report
Comments and Suggestions for Authors
This short narrative review examines the function of artificial intelligence (AI) in improving the effectiveness of cancer immunotherapy, forecasting patient responses, and identifying new therapeutic targets. Approaches: A thorough examination of the literature was performed, concentrating on studies published from 2010 to 2024 that investigated the utilisation of AI in cancer immunotherapy. The paper discusses an important and hot topic, but it lacks statistics, and explanation or even mentions what machine/deep learning techniques are used in cancer immunotherapy. Some comments need to be addressed to the quality of the paper.
What differentiates this review from the existing literature on AI in cancer immunotherapy?
Please highlight the novelty of the review, aims, and objectives of the review in the form of points.
Please discuss Figures 1 and 2 in the text.
Please provide details regarding how many papers were initially selected, how many papers were omitted and the final number of papers involved in the review.
Please discuss the findings in Table 1 in the text.
The paper lacks some discussion on the machine and deep learning techniques used in cancer immunotherapy. Please add the techniques used.
Please add some more figures that show statistics regarding how many papers were used in each area or subsection of the review. Also, there could be bar or pie charts showing the distribution of machine/deep learning techniques among different areas included in the review.
Please add limitations and challenges into a dedicated section.
This short narrative review examines the function of artificial intelligence (AI) in improving the effectiveness of cancer immunotherapy, forecasting patient responses, and identifying new therapeutic targets. Approaches: A thorough examination of the literature was performed, concentrating on studies published from 2010 to 2024 that investigated the utilisation of AI in cancer immunotherapy. The paper discusses an important and hot topic, but it lacks statistics, and explanation or even mentions what machine/deep learning techniques are used in cancer immunotherapy. Some comments need to be addressed to the quality of the paper.
What differentiates this review from the existing literature on AI in cancer immunotherapy?
Please highlight the novelty of the review, aims, and objectives of the review in the form of points.
Please discuss Figures 1 and 2 in the text.
Please provide details regarding how many papers were initially selected, how many papers were omitted and the final number of papers involved in the review.
Please discuss the findings in Table 1 in the text.
The paper lacks some discussion on the machine and deep learning techniques used in cancer immunotherapy. Please add the techniques used.
Please add some more figures that show statistics regarding how many papers were used in each area or subsection of the review. Also, there could be bar or pie charts showing the distribution of machine/deep learning techniques among different areas included in the review.
Please add limitations and challenges into a dedicated section.
Author Response
Reviewer 3:
This short narrative review examines the function of artificial intelligence (AI) in improving the effectiveness of cancer immunotherapy, forecasting patient responses, and identifying new therapeutic targets. Approaches: A thorough examination of the literature was performed, concentrating on studies published from 2010 to 2024 that investigated the utilisation of AI in cancer immunotherapy. The paper discusses an important and hot topic, but it lacks statistics, and explanation or even mentions what machine/deep learning techniques are used in cancer immunotherapy. Some comments need to be addressed to the quality of the paper.
What differentiates this review from the existing literature on AI in cancer immunotherapy?
Response: We appreciate your observation. To address this, we have expanded the introduction to highlight what differentiates this review (lines 124 – 163). While previous studies focus on isolated applications of AI in cancer immunotherapy, this review uniquely synthesizes these aspects while addressing challenges such as algorithm transparency, regulatory pathways, and real-world implementation. The review is positioned within the broader field to provide actionable insights for researchers, clinicians, and policymakers, making it relevant to advancing precision medicine.
Please highlight the novelty of the review, aims, and objectives of the review in the form of points.
Response: We have included a clear statement of the review’s novelty, aims, and objectives at the end of the introduction (lines 132 – 151). Specifically, the review is novel in its integration of AI’s current contributions with practical challenges and forward-looking perspectives. The key objectives are: (1) to summarize the current applications of AI in enhancing the efficacy of cancer immunotherapy, (2) to identify challenges in integrating AI into clinical practice, and (3) to propose future research directions. These additions clarify the unique contribution and scope of the review.
Please discuss Figures 1 and 2 in the text.
Response: We have incorporated a concise discussion of Figures 1 and 5 (previous figure 2) into the text to better align them with the manuscript.
Figure 1 (lines 152 – 163) illustrates the multi-step process of integrating AI into cancer immunotherapy, from data acquisition (genomic sequences, medical records, immune profiling, imaging) to AI/ML model training, which predicts patient responses and personalizes treatments. It highlights a feedback loop, where patient progress informs continuous improvement in care delivery. Figure 5 (lines 312 – 321) focuses on biomarker discovery, detailing how genomic, transcriptomic, and proteomic data are analyzed by AI to identify biomarkers like PD-1, TMB, and MSI for guiding personalized clinical decisions. These discussions contextualize the figures, emphasizing their relevance in demonstrating AI’s transformative role in cancer immunotherapy. Thank you for helping us enhance the clarity of the manuscript.
Please provide details regarding how many papers were initially selected, how many papers were omitted and the final number of papers involved in the review.
Response: Thank you for your comment. In response, we have included detailed information about the study selection process (lines 184 – 189). Initially, 360 articles were identified across four databases: PubMed (120), Google Scholar (80), Scopus (100), and Web of Science (60). After screening, 268 articles were excluded based on predefined inclusion and exclusion criteria, leaving 92 articles for full-text review. Following a thorough evaluation, an additional 38 articles were excluded, resulting in 54 studies included in the final review. These details are now explicitly stated in both the methodology section and represented visually in Figure 2 through the PRISMA flowchart.
Please discuss the findings in Table 1 in the text.
Response: We have expanded the text to thoroughly discuss the findings presented in Table 1, ensuring clear connections between the AI applications, methodologies, and their impacts on cancer immunotherapy (lines 279 – 284). For example, in biomarker identification, the text now elaborates on the use of Random Forest (RF), Support Vector Machines (SVM), and Convolutional Neural Networks (CNNs) to analyze multi-omics data for identifying biomarkers like TMB, MSI, and PD-L1 expression levels. Similarly, therapeutic target discovery is discussed with specific reference to Generative Adversarial Networks (GANs) and clustering methods for uncovering neoantigens and novel immune checkpoints, expanding therapeutic options (lines 422 – 430).
The text also highlights the use of Long Short-Term Memory (LSTM) networks and time-series analysis for real-time monitoring, enabling the timely detection of immune-related adverse events (lines 526 – 532). Each application aligns directly with the table, ensuring a cohesive narrative that links the methodologies to their clinical impacts, such as improving patient safety, accelerating drug discovery, and optimizing personalized treatments. This integration ensures consistency between the table and the main text while providing a comprehensive overview of AI’s transformative role in cancer immunotherapy.
The paper lacks some discussion on the machine and deep learning techniques used in cancer immunotherapy. Please add the techniques used.
Response: We have revised Table 1 to include specific machine learning (ML) and deep learning (DL) techniques used in cancer immunotherapy. These updates detail algorithms such as Random Forest, Support Vector Machines (SVM), Reinforcement Learning, Generative Adversarial Networks (GANs), and Long Short-Term Memory (LSTM) networks, providing clarity on the methodologies applied. Additionally, we have ensured that these techniques have been discussed Section 4 of the manuscript to complement the table. This enhancement offers a more comprehensive overview of how specific AI models are leveraged to address challenges in cancer immunotherapy.
Please add some more figures that show statistics regarding how many papers were used in each area or subsection of the review. Also, there could be bar or pie charts showing the distribution of machine/deep learning techniques among different areas included in the review.
Response: Thank you for your comment. In response, we have added Figure 3, titled "Distribution of Papers Across Key Subsections of the Review," which provides a statistical overview of how the 54 studies included in the review are distributed across various areas. The figure highlights the number of studies focusing on biomarker identification, predicting patient response, drug development, combination therapies, CAR-T cell therapy, monitoring treatment response, and clinical trial optimization. This visual representation enhances the understanding of the evidence base supporting each subsection, aligning with your suggestion to include statistical insights.
Also, we have added Figure 4, titled "Distribution of Machine and Deep Learning Techniques Across Review Areas," to visually represent the application of these techniques in various aspects of the review. The figure highlights that biomarker identification accounts for the largest proportion of machine and deep learning applications, followed by predicting patient response and drug development. Other areas, such as combination therapies, CAR-T cell therapy, monitoring treatment response, and clinical trial optimization, are also well-represented.
Please add limitations and challenges into a dedicated section.
Response: we have added a dedicated Limitations and Challenges section addressing barriers to integrating AI into cancer immunotherapy (lines 688 – 742). Key issues include the need for high-quality, diverse datasets, the "black box" nature of many AI models limiting transparency and trust, and regulatory hurdles such as compliance with FDA and CE standards and the need for resource-intensive clinical trials. Logistical challenges, including interoperability with healthcare systems and clinician training, are also discussed. Ethical concerns around data privacy, consent, and accountability are highlighted as critical barriers. This section balances AI’s potential with practical challenges, ensuring a comprehensive discussion. Thank you for helping us enhance the manuscript.
Reviewer 4 Report
Comments and Suggestions for Authors
In the present paper auhors provide a review on the role of artificial intelligence in enhancing the efficacy of cancer immunotherapy, predicting patient responses, and discovering novel therapeutic targets.
The manuscript is well written and data are clearly presented. To improve the overall quality i have the following suggestions:
- Pathological evaluation of neoplastic tissues is a key step in determining patients eligibility to immunoptherapy. To date, severals studies have been published on this topic (PD-L1, Mismatch repair immunhistochemical evaluation with AI models; TILS count with AI models). Additionally, AI tools are able to predict the MMR status or susceptibility to ICI therapy on the basis of the morphological features observed in haematoxylin and eosin stained histological sections.
Authors should include a few additional paragraphs to discuss the latest research on these topics.
Author Response
Reviewer 4:
In the present paper authors provide a review on the role of artificial intelligence in enhancing the efficacy of cancer immunotherapy, predicting patient responses, and discovering novel therapeutic targets.
The manuscript is well written and data are clearly presented. To improve the overall quality i have the following suggestions:
- Pathological evaluation of neoplastic tissues is a key step in determining patients eligibility to immunoptherapy. To date, severals studies have been published on this topic (PD-L1, Mismatch repair immunhistochemical evaluation with AI models; TILS count with AI models). Additionally, AI tools are able to predict the MMR status or susceptibility to ICI therapy on the basis of the morphological features observed in haematoxylin and eosin stained histological sections.
Authors should include a few additional paragraphs to discuss the latest research on these topics.
Response: We have expanded the section 5.3 to include a detailed discussion of the latest research on AI’s role in pathological evaluation, particularly in assessing mismatch repair (MMR) deficiency and other key biomarkers (lines 608 – 635). The revised section highlights AI’s ability to automate PD-L1 expression evaluations and tumor-infiltrating lymphocyte (TIL) quantification, both critical for determining immunotherapy eligibility. Additionally, we have included insights into AI’s capability to predict MMR status and susceptibility to immune checkpoint inhibitor (ICI) therapy by analyzing morphological features from hematoxylin and eosin (H&E)-stained slides. These advancements underscore AI’s transformative potential in enhancing diagnostic precision and optimizing patient selection for immunotherapy.
Round 2
Reviewer 1 Report
Comments and Suggestions for Authors
The authors have conscientiously addressed the comments and made edits to my and other reviewers' suggestions. This review is of a much better quality than its previous version. I believe that this work will provide a good context for AI and DL methods for cancer immunology from detection, diagnosis and patient care.
Reviewer 2 Report
Comments and Suggestions for Authors
Congratulations
Reviewer 3 Report
Comments and Suggestions for Authors
The authors have addressed all my comments